# STAM: Zero-Shot Style Transfer using Diffusion Model via Attention Modulation

## Abstract

*Diffusion models serve as the basis of several different zero-shot image editing applications, including image generation and style transfer. The basic approach in style transfer using diffusion models involves swapping attention components between the provided content and style images. Straightforward interchange of these components can lead to inadequate style injection or loss of content image characteristics. This paper addresses shortcomings of attention-guided style transfer by two novel contributions: a) preserving content via dual path attention aggregation and b) maintaining the impact of style through modulation of attention components. The proposed STAM approach can provide aesthetically appealing yet content-preserving style transfer through a combination of these contributions and is also applicable to prompt-driven style transfer. STAM is validated by extensive qualitative and quantitative evaluations and compared to ten recent works that are largely outperformed by the proposed work. In addition to style transfer quality, STAM is also compared to previous work in terms of inference time and remains close to the fastest competing approaches.*

## 1. Introduction

Text-to-Image (T2I) diffusion models [19, 40] offer excellent image generation performance from random noises with respect to the given prompts, and sometimes it is hard to distinguish them from natural images. In addition to image generation, these models also show great promise in several zero-shot image editing tasks [2, 4, 15]; however, for recent T2I models, style transfer still poses a considerable challenge.

In the image style transfer task, a fundamental trade-off exists between preserving the content of the source image and effectively injecting style details from the reference image. Despite advancements, style transfer methods continue to face challenges in balancing this tradeoff. Techniques like neural style [13, 21] often prioritize style

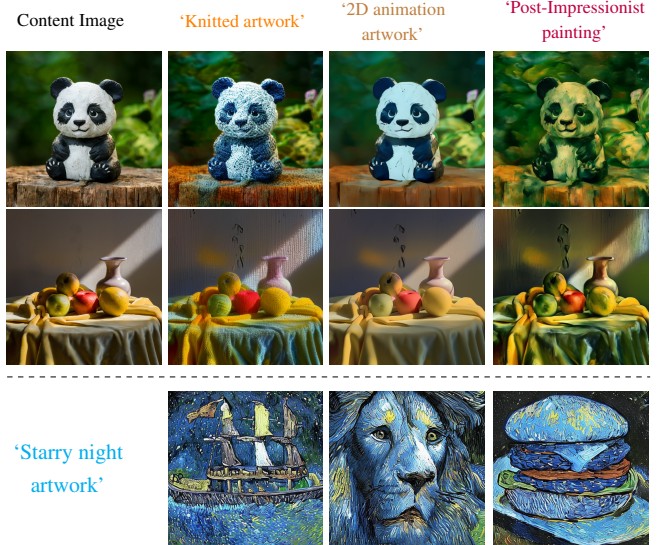

Figure 1. Given the content image, our method successfully reflects the true intent of the prompt in the stylized image. Additionally, for a given prompt, our approach can successfully generate style-aligned images..

at the expense of content fidelity, while [46, 48] struggle to fully retain structural integrity when enhancing stylistic attributes like texture and color. In existing diffusion-based approaches [8, 10, 14, 22], stylized image reconstruction requires the query from the source image to preserve the content information. For style injection, key and value information comes from the style image. In this process, the stylized image inherits a swapped attention mechanism for the desired reconstruction while taking guidance from the source and style images. However, this basic self-attention swapping approach struggles to address the content preservation vs. style intensity tradeoff [10], and essentially necessitates further attention *modification* for a desirable outcome [10, 14, 23].

This study proposes a novel zero-shot approach, Style Transfer with Attention Modulation (STAM), which ad-

dresses the above concerns via attention modulation and aggregation. We leverage a pre-trained diffusion model, applying DDIM [42] inversion to project content and style images into latent space and initialize the stylized output with the content's latent copy. We propose two novel approaches to jointly address the aforementioned tradeoff: a) *Attention Modulation* for style regulation, and b) *Attention aggregation* for content preservation. The proposed attention modulation is a simple technique that adjusts the attention components, query, key, and value according to the user's choice via hyperparameter selection. The proposed attention aggregation approach blends attention pathways from content and style queries to regulate their influence, ensuring content preservation while enhancing style application effect. This dual strategy, applied within a U-Net's attention layers, achieves a harmonious content-style balance, offering customizable styling without the computational burden of fine-tuning, as demonstrated in our results.

Overall, STAM is a novel style transfer approach that relies on *attention modulation* and *attention aggregation* to deliver superior stylization performance while preserving the content information with user control. We summarize the contributions of STAM below:

- A novel zero-shot approach for stylizing natural images with a fine control between style regulation and content preservation.

- Our stylization algorithm uses a novel attention modulation that allows style regulation without additional latent tuning and preserves the content information via a novel attention aggregation scheme.

- Our approach is free of additional adapters and can address prompt-image misalignment.

## 2. Related Works

### 2.1. Text-to-Image generation

Text-to-image generation (T2I) has achieved unprecedented advancements, driven by the successive evolution of diffusion models [12, 34]. These models [31, 35, 38] harness incredible capabilities in high-quality image generation and synthesis from textual descriptions, broadening the boundaries of applications such as inpainting [1, 37], image-to-image translation [45, 51], local image editing [9], and video generation [24].

### 2.2. Diffusion-based style transfer

Diffusion-based style transfer approaches can be categorized into two distinct methodologies: training or fine-tuning-based approaches and zero-shot approaches. Each differs in its reliance on model adaptation for style application.

**Training based approaches.** Earlier style transfer approaches leverage weak supervision on curated datasets like WikiArt [28], using CLIP-based losses [32] to enhance style representation, as seen in DiffStyler [26] with aesthetic loss, StyleDiffusion [48] with disentanglement loss, and Dream Inpainter [49] for conditional generation, while SmartBrush adds segmentation masks and Instruct Pix2Pix [3] trains fully supervised models with textual instructions. Methods like [11, 21, 30] employ adaptive normalization and transformers for visual enhancement.

In contrast, fine-tuning-based approaches adapt pre-trained diffusion models by adjusting parameters [7, 44] or optimizing text/null-text embeddings [29], with StyleDiffusion [48] mapping features to cross-attention layers, InST [54] refining text-image interplay via multilayer cross-attention, and others [20, 36, 41] learning styles from reference images, though CSGO's unified framework [50] requires extensive training, limiting adaptability to unseen data or single references.

**Zero-shot approaches.** These training- and fine-tuning-free approaches can be categorized into two groups: (i) altering attention mechanism features or values to maintain spatial alignment between generated images and target styles [8, 11, 23], and (ii) employing region-based editing with segmentation masks [1, 9]. Prompt-2-Prompt (P2P) [16] utilizes cross-attention layers to adjust image layouts to textual prompts, while MasaCtrl [5] and PnP [43] refine attention value computations for better stylization control. However, these methods struggle with contextual fidelity—e.g., StyleAligned [17] distorts content—prompting InstantStyle [46] to integrate pre-trained networks [51, 52] for consistency. ControlNet [47] enhances inversion with geometric constraints like depth maps and sketches, though its reliance on priors limits flexibility, whereas Artist [23] achieves harmonic stylization via content-style disentanglement without structural networks.

## 3. Methodology

The proposed STAM approach utilizes a pre-trained diffusion model to concurrently sample the content image, style image, and stylized content image. Before providing the content image to the diffusion model, DDIM inversion is applied to project the content image to the preferred latent space. The same is done for the style image if one is available; otherwise, the style prompt is provided with a random latent. For the stylized output, the inverted latent is copied from the content image as the initial input for the diffusion model. During sampling, in the attention layers, attention decoupling is performed between the content, style, and stylized content to obtain the desired outcome. Such so-called attention decoupling is popular among the recent diffusion-based style transfer approaches. Before unveiling our proposed attention decoupling approach, we describe

essential preliminary concepts and challenges among regular attention decoupling approaches.

## 3.1. Preliminaries

We present several preliminary concepts relevant to the STAM: diffusion, DDIM inversion, and self-attention.

**Latent diffusion models.** Diffusion models operate in latent space to mitigate the dimensional complexity of pixel-space processing. A pre-trained auto-encoder maps the user-provided input into an initial latent representation, denoted $\mathbf{z}_0$. During the forward diffusion process, this latent $\mathbf{z}_0$ undergoes a Markov transition, progressively incorporating Gaussian noise until it evolves into the latent state $\mathbf{z}_t$ at time step $t$. The probability density of $\mathbf{z}_t$ conditioned on $\mathbf{z}_{t-1}$ is given by:

$$q(\mathbf{z}_t|\mathbf{z}_{t-1}) = \mathcal{N}(\mathbf{z}_t; \sqrt{1-\beta_t}\mathbf{z}_{t-1}, \beta_t\mathbf{I}), \qquad (1)$$

In the above, $\beta_t$ represents the variance schedule, where $t$ denotes a specific timestep, and $T$ signifies the total number of time-steps in the training process. In the backward phase, a denoiser U-Net denoted $\epsilon_\theta$, is employed, expressed as follows:

$$p_\theta(\mathbf{z}_{t-1}|\mathbf{z}_t) = \mathcal{N}(\mathbf{z}_{t-1}; \mu_\theta(\mathbf{z}_t, \tau, t), \Sigma_\theta(\mathbf{z}_t, \tau, t)), \quad (2)$$

Above, $\tau$ is the textual prompt for the given input. To compute the mean $\mu_\theta$ and variance $\Sigma_\theta$, we use the U-Net $\epsilon_\theta$.

**DDIM inversion.** Through DDIM sampling [42], it is possible to deterministically transform a random noise latent $\mathbf{z}_t$ into a denoised latent $\mathbf{z}_0$ by following a timestep sequence $t : T \to 1$. The DDIM sampling equation can be formulated as follows:

$$\mathbf{z}_{t-1} = \sqrt{\alpha_{t-1}} \frac{\mathbf{z}_t - \sqrt{1-\alpha_t}\epsilon_\theta}{\sqrt{\alpha_t}} + \sqrt{1-\alpha_{t-1}}\epsilon_\theta, \quad (3)$$

Drawing from the ODE analysis of the diffusion process, deterministic DDIM inversion [42] enables the projection of a clean latent $\mathbf{z}_0$ into a noisy latent $\hat{\mathbf{z}}_T$. In this case, the time sequence proceeds in reverse order, $t : 1 \to T$:

$$\hat{\mathbf{z}}_t = \sqrt{\alpha_t} \frac{\hat{z}_{t-1} - \sqrt{1-\alpha_{t-1}}\epsilon_\theta}{\sqrt{\alpha_{t-1}}} + \sqrt{1-\alpha_t}\epsilon_\theta. \quad (4)$$

In the equation above, $\alpha_t$ denotes the noise scheduling parameter [18, 42].

**Self-attention.** The attention mechanism first introduced groundbreaking improvement in machine translation, followed by similar impacts in other domains. In essence, the self-attention mechanism projects the input latent into a query $Q$, a key $K$, and a value $V$ and performs the following operation:

$$Self\_Attn(Q, K, V) = Softmax(\frac{Q, K^T}{\sqrt{d}}) \cdot V \qquad (5)$$

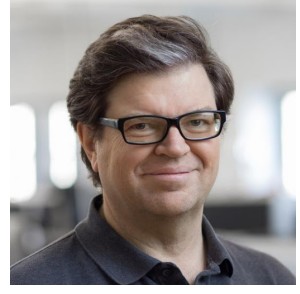 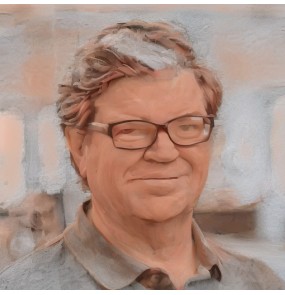

(a) Content Image    (b) Stylized Output

Figure 2. Outcome of basic self-attention swapping during style transfer. For the style prompt *terracotta style*, it can be seen that stylized output only captures the red clay color within the layout.

**Swapping self-attention.** Considering three images: Content ($c$), Reference ($r$), and Stylized ($s$), we can denote the attention components as follows: $Q_c, K_c, V_c$, $Q_r, K_r, V_r$, and $Q_s, K_s, V_s$. In regular diffusion sampling, the stylized image is created by incorporating so-called *attention decoupling* [10]. In practice, attention decoupling involves swapping the attention components among the content, reference, and stylized images. In attention decoupling, the query of the stylized image is swapped with the query from the content image. Also the key and value of the stylized image are swapped with the key and value of the reference image. In terms of an equation, this can be written as follows:

$$Self\_Attn(Q_s, K_s, V_s) = Softmax(\frac{Q_c, K_r^T}{\sqrt{d}}) \cdot V_r \qquad (6)$$

The straightforward approach of swapping has a tradeoff between maintaining style patterns and preserving content. It also necessitates careful selection of layers for the specified UNet; otherwise, content from the reference image may leak into the output. Additionally, in our work we have observed that this naive method captures style information to the output only partially. For instance, Figure 2 illustrates a case where only certain areas of the image adhere to the provided style prompt.

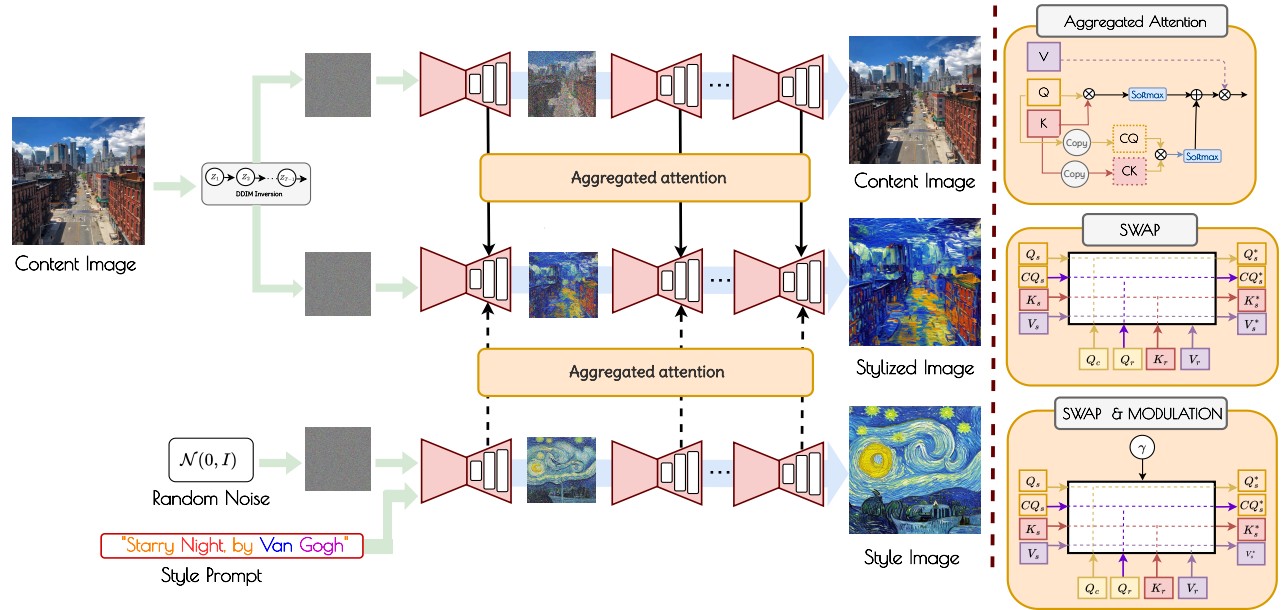

Figure 3. The flowchart of the proposed STAM style transfer approach. On the rightmost part of the diagram, three sub-blocks are shown with the underlying mechanics of the proposed aggregated attention, attention swapping, and modulation of the attention components. We do not show the full details of attention modulation here for brevity. $Q_s$ is denoted as $Q_s^*$ to indicate the completion of the appropriate transformation.

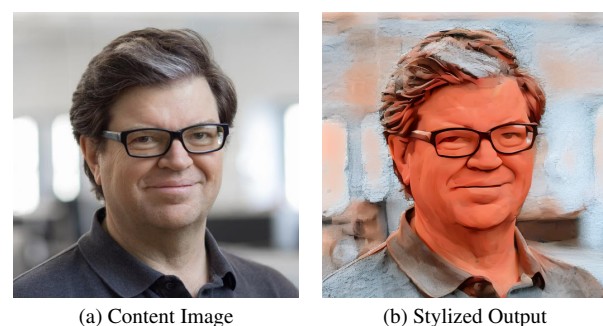

(a) Content Image          (b) Stylized Output

Figure 4. Attention modulation [10] applied on top of self-attention decoupling as in [22]. The resultant image conveys the terracotta style with enhanced contrast only. This observation signifies that modifying the attention matrix may not always bring the desired text-image alignment.

This lack of matching the intended style from the given prompt or style depends on the complexity of the given style image or prompt. One particular way of addressing it is attention modulation [10]. There are many ways to modulate the attention matrices, and we observed that modulation can improve the contrast or intensity of the style strokes, but faithful style injection has still been missing. We also present another example, where we have applied attention modulation [10], resulting in somewhat more visually appealing outcomes than before but lacking proper

style injection, as shown in Figure 4. It is also evident that directly modulating [10] may not always guarantee the desired prompt effect in the stylized output. To address these shortcomings, we propose a two-step solution: a) attention modulation and b) attention aggregation. We describe them briefly below.

**Attention modulation.** Even though available diffusion models are great in image generation, they suffer from insufficient alignment between the prompt and the output. Such phenomena can be observed in diffusion-based zero-shot tasks as well. To address this, researchers have used attention modulation, such as on-the-fly latent tuning [6,25,27,39], to enforce attention refinement. However, latent tuning increases the sampling time by multiple folds, which calls for pursuing more efficient approaches. However, a fundamental trend can be observed in the available attention modulation approaches: modulating the attention after attention matrix computation. Using this approach, the impact of the *prompt* is already conveyed in the attention matrix, affecting the concurrent latent tuning quality.

Intuitively, attention modulation is necessary to enable proper prompt-image alignment during the stylization process, but directly modulating the attention matrix sometimes performs worse than expected (See Figure 4). Instead, if we modulate the query, key, and value, we might have a better cosine similarity response within the attention

matrix. However, tuning these attention components during sampling is cumbersome, and this scenario is further worsened by the number of cross-attention layers from the diffusion model. Hence, we propose modulating them via gamma correction; in this way, it is possible to avoid latent tuning or attention component tuning. Additionally, it gives the user more editing flexibility.

To perform gamma correction, we normalize the attention components to a range of 0 to 1, apply modulation using a selected gamma value, and then rescale them back to their original ranges. Denoting $\mathcal{T}_n()$ as the normalization function and $\mathcal{T}_d()$ as the rescaling function, the attention modulation process can be formulated as follows:

$$Q^m = \mathcal{T}_d(\mathcal{T}_n(Q)^{\gamma_1}), K^m = \mathcal{T}_d(\mathcal{T}_n(K)^{\gamma_2}), V^m = \mathcal{T}_d(\mathcal{T}_n(V)^{\gamma_3}) \quad (7)$$

Here, $\gamma_1$, $\gamma_2$, and $\gamma_3$ are user-specific values, and $Q^m, K^m, V^m$ denote modulated attention components. By applying this attention modulation, we can address the lack of style injection within the output image and regulate the intensity of the style information. To support this claim, the necessary ablations are presented later in the paper. However, whereas this modulation might be tuning-free, it can unfortunately overlook the content information, introducing the style vs. content tradeoff. To emphasize content preservation, we propose *attention aggregation* via dividing the attention path as detailed below.

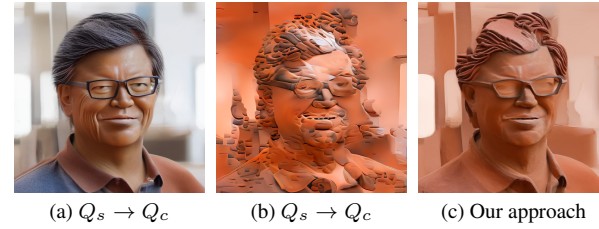

(a) $Q_s \rightarrow Q_c$     (b) $Q_s \rightarrow Q_c$     (c) Our approach

Figure 5. Here, we display the outcome of different attention choices via swapping attention components: a) Replacing $Q_s$ with $Q_c$, b) Replacing $Q_s$ with $Q_r$, and c) our dual-path attention aggregation approach. The visual comparison shows that our approach can faithfully preserve the content layout while sufficiently containing style information.

**Aggregated attention.** We propose a novel attention aggregation strategy to improve content preservation during style transfer. Our aggregation policy follows an empirical observation within the swapping attention process. In the naive attention-swapping scheme, we replace the $Q_s, K_s, V_s$ with $Q_c, K_s, V_s$, which slightly captures the style strokes but mostly preserves the content. If we replace $Q_s, K_s, V_s$ with $Q_r, K_s, V_s$, an overflow of style can be observed in the outcome, as shown in Figure 5. Through our

aforementioned modulation scheme, we regulate the impact of the content and style from the attention-swapping pathways. Based on these observations, the most promising combination of swapping alternatives can be discovered intuitively: a copy of $Q_s, K_s, V_s$ is instantiated as $CQ_s, CK_s, CV_s$, followed by different attention swaps applied to the copies together with computing attention matrices and aggregations between them. Hence, the attention for the $Q_s, K_s$ is:

$$A_s = Softmax(\frac{Q_c, K_s{}^T}{\sqrt{d}}) \quad (8)$$

Similarly, computing attention for the $CQ_s, CK_s$ as:

$$CA_s = Softmax(\frac{CQ_r, K_s{}^T}{\sqrt{d}}) \quad (9)$$

Then, finally, we formulate the aggregated attention as follows:

$$Ag\_Attn(Q_s, K_s, V_s) = 0.5 * (CA_s + A_s) \cdot V_s \quad (10)$$

As a result of these mixed attention pathways, the approach STAM approach provides a satisfactory balance between content information preservation and style application, as displayed in Figure 5. Additionally, the modulation technique used on the attention components gives us an extra performance boost with a customizable stylization experience. The following sections include the necessary results for the style transfer and ablations for the modulation parameters.

## 4. Experiments

### 4.1. Experimental Setup

**Implementation details.** We use the pre-trained Stable Diffusion 2.1 model for our main experiments, and perform DDIM sampling with $T = 50$ steps. During the inversion phase, we recorded the intermediate noise predictions, which are further utilized to overwrite the input of the denoiser of the content image at every step. We set the scale of the classifier-free guidance as 7.5. Our experiments are conducted on a single RTX 3090 GPU server.

**Baseline methods.** We subdivided our comparison procedure into two different sections. **1) text-driven methods and 2) reference-style image-based methods**. The first section shows a comparison with different text-driven and general manipulation methods: DDIM inversion [42], ControlNet [52], InstructPix2Pix [3], Diffartist [23], and CSGO [50]. The second section consists of a comparison between different reference style image-based methods and ours. For this experiment, we generate style images from the target

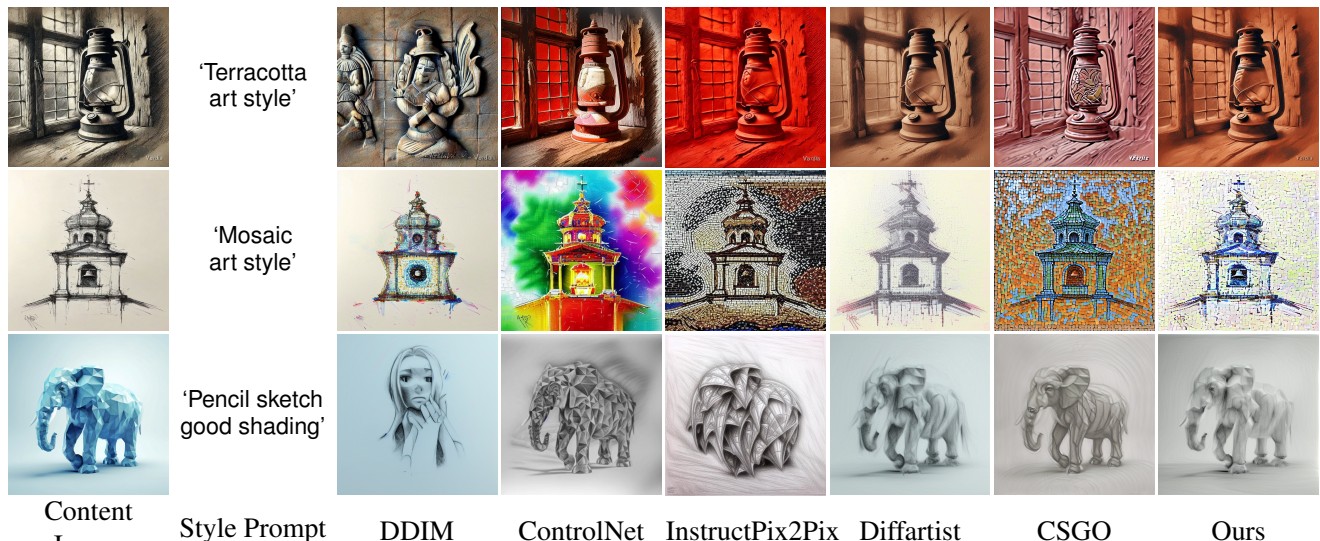

Figure 6. A qualitative comparison of text-driven style transfer, where the comparison includes DDIM [42], ControlNet [52], Instruct-Pix2Pix [3], Diffartist [23], CSGO [50], and the proposed STAM. The proposed method effectively captures the desired artistic characteristics while maintaining high fidelity and contextual coherence.

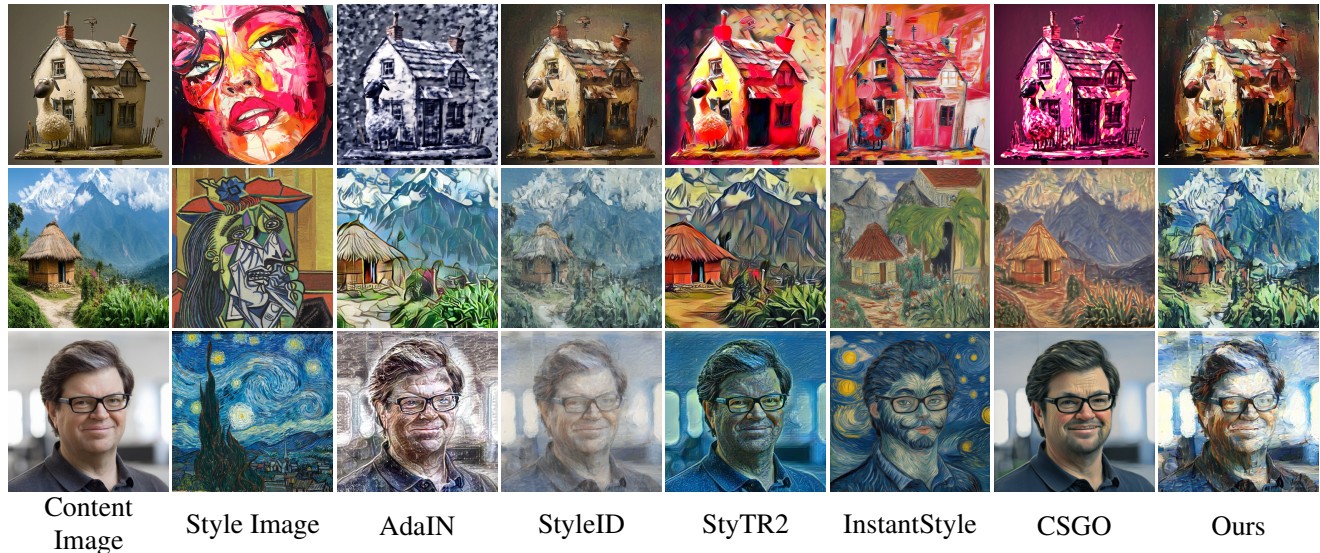

Figure 7. Qualitative comparison with image-driven style transfer methods, where the comparison includes AdaIN [21], Style ID [8], StyTR2 [11], Instant Style [46], and CSGO [50]. Our method maintains the identity of objects, landscapes and faces while avoiding excessive distortions and unnatural artifacts, resulting in a more coherent and visually appealing stylization.

prompts set as the reference style image. The compared methods for this subsection are: AdaIN [21], StyleTr2 [11], StyleID [8], InstantStyle [46], Artist [23], and CSGO [50].

**Performance Metrics.** We use LPIPS [53] to assess the content similarity between the original content image and the stylized output. Meanwhile, to evaluate the style similarity, we employ CLIP alignment [33] , which measures the correspondence between the generated stylized image

and the reference style image.

## 4.2. Experimental Results

**Comparison with text-driven approaches.** Figure 6 presents a qualitative comparison with several text-driven image-based approaches, demonstrating the competitive performance of STAM in style transformation to content images according to specific artistic style prompts. By observing both *hurricane* and *dome* stylization, our method

| Metric | Ours | DDIM | SD | PnP | P2P (w/o NTI) | InstructPix2Pix | ControlNet | InstantStyle | DiffArtist | CSGO |
|---|---|---|---|---|---|---|---|---|---|---|
| Inference speed(s) ↓ | 10.7 | 9.7 | 3.9 | 55.3 | 29.1 | 9.2 | 7.8 | 7.8 | 10.7 | 18.4 |
| LPIPS ↓ | 0.43 | 0.57 | 0.76 | 0.63 | 0.48 | **0.40** | 0.65 | 0.59 | 0.51 | 0.47 |
| CLIP Alignment↑ | **27.91** | 25.98 | 27.17 | 27.74 | 23.48 | 21.00 | 25.20 | 22.33 | 24.39 | 27.41 |

Table 1. Quantitative comparison of the proposed method against existing approaches. This evaluation includes training-associated and training-free methods, assessing the performance of each method for inference time, content reservation, and style quality.

effectively captures the structural layout of the original content while accurately reflecting both terracotta and mosaic art styles. For stylization of the *dome*, our method effectively captures the traditional mosaic art style, ensuring natural and coherent *mosaic patch* transitions. Similarly, *elephant* stylization experiences an optimal combination of *low-poly* style from the content and *pencil sketch* outlook from the given text prompt, preserving sharp edges and structural folds while taking a sketch artwork form. In comparison, both DiffArtist [23] and CSGO [50] introduce several distortions, with Diffartist failing to incorporate the original discriminative textures and CSGO producing unnatural pixelated effects that can disrupt stylistic harmony.

**Comparison with image-driven approaches.** Figure 7 shows a style-image-driven comparison. The three different visual contents undergo well-structured composition: successful retention of the content details while effectively transferring the color palette and texture from the style image. In contrast, methods such as CSGO [50] introduce extreme color distortions and unnatural overlays with high-contrast effects. Methods such as StyleID [8] and StyTR2 [11] exhibit various degrees of abstraction but face an imbalance in artistic stylization with content preservation compared to our approach.

**Quantitative Comparisons.** Table 1 presents a quantitative comparison between our proposed text-driven stylization method and existing style transfer techniques. In this evaluation, we specifically consider the hybrid method CSGO, which is used exclusively for text-driven stylization. The experimental setup consists of 40 diverse style prompts sampled from the WikiArt dataset [28], 50 content images from the PASCAL-VOC dataset, and 50 photorealistic images generated using various diffusion models.

To assess computational efficiency, we conduct a basic inference speed comparison. Our method demonstrates performance comparable to Diffartist [23] in terms of computational complexity. Regarding performance metrics, our approach achieves the second-best overall ranking, with InstructPix2Pix [3] obtaining the highest score. Furthermore, we evaluate the text-driven stylized synthesis using the CLIP alignment metric, where our method outperforms

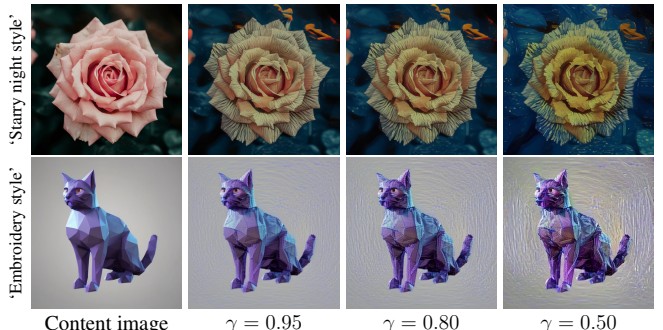

Figure 8. Ablation study on the effect of the parameter $\gamma$. We vary $\gamma$ for the key component, observing that a lower value of $\gamma$ results in higher style intensity, demonstrating the impact of $\gamma$ in controlling the balance between content preservation and style adaptation.

Stable Diffusion (SD), achieving the highest score. A particularly noteworthy finding is the superior integration of style and content achieved by our method. When analyzing the CA-Style and SA-Content scores, our approach attains the best results, underscoring the effectiveness of our technique in maintaining a harmonious balance between content and style.

## 5. Ablations

In this section, we present necessary discussions on the effect of the modulation parameters, style-driven image generation, and style injection vs content preservation ablation.

**Impact of the modulation parameters.** The AM method of STAM modulates the attention components (query, key, and value) that refine the attention outcome without any tuning cost and return more prompt-aligned output. Figure 8 shows the impact of the $\gamma$ parameter during attention modulation. For this instance, we only modulate the Key with the varying value of $\gamma$ and keep $\gamma = 1$ for the other two components. We start by assigning the prompt as *Starry night style* and fixing $\gamma = 0.95$. In the case of Figure 8, we see that rose petals get a pale-yellowish color with brush stroke marks, indicating that the intended prompt-image alignment

is lacking. Next, we perform the style transfer again, changing $\gamma = 0.80$, and the stylized outcome is now spatially more aligned with the *Starry night* look, as yellow stars appear in the background of the rose image. Finally, setting $\gamma = 0.50$ provides a pleasant output, where we can find prominent features of the *starry night* artwork, such as blue-yellow color dominance, dotted stars, or swirling brush strokes within our stylized image. We also present another example where the input image contains a cat with a low-poly form, and we stylize it with the *embroidery* look, with identical parameter choice from the previous example. Like before, we also found the same style intensity variation here for the chosen $\gamma$ parameter.

*'Pencil sketch art'*

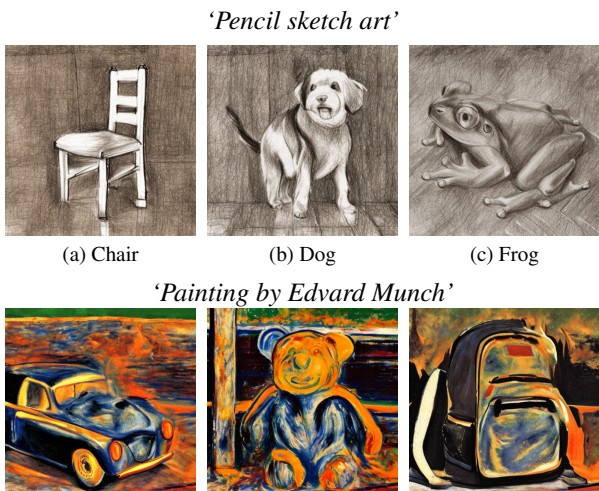

(a) Chair      (b) Dog      (c) Frog

*'Painting by Edvard Munch'*

(d) Toy car      (e) Teddy bear      (f) School bag

Figure 9. Our STAM is capable in utilizing T2I models for text-driven style-aligned subject generation as well.

**Style-aligned subject generation.** STAM can also utilize T2I models for generating images with identical styles. For a given prompt, our approach can direct the given diffusion model to generate images fashioned by the input style prompt. In Figure 9, we present two examples where STAM generates multiple subjects based on the input style prompts.

**Style injection vs.content preservation.** In Figure 10, we demonstrate a qualitative comparison for style injection vs. content preservation tradeoff between [50], [8], and our study. Our experience shows that StyleID [8] excels in preserving the content layout but is not consistent in style injection. CSGO [50] can clean the content image with the extracted style at the cost of content fidelity. From Figure 10, we see that CSGO [50] captures the color tone adequately from the given style image; however, it changes the layout

of the person severely, whereas StyleID [8] does the opposite. In contrast to them, STAM offers an optimal balance between content preservation and adequate style injection for the given image. The same observation is also valid for the lion image, where our method can successfully transfer the style in the output image without severely changing the lion image appearance.

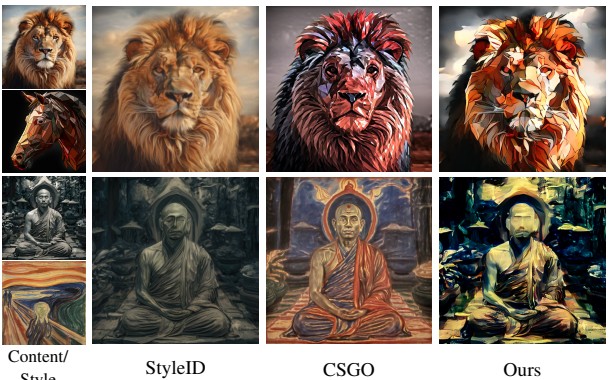

Content/    StyleID    CSGO    Ours
Style

Figure 10. Style transfer vs. content preservation tradeoff for CSGO [50], StyleID [8], and our study. Compared to [50] and [8], our approach delivers balanced content preservation and style injection, resulting in a visually pleasant outcome.

## 6. Conclusion

We propose STAM, a zero-shot style transfer approach that leverages pre-trained T2I diffusion models. Our study addresses two tradeoffs in the standard style transfer algorithms: a) regulation of style injection and b) content preservation. We regulate the intensity of the given style by modulating the query, key, and value of the given content-style pair. To preserve the content information in the stylized image, we rely on our proposed attention aggregation mechanism, which mixes the style and content information during cross-attention. By joint effort of these novel contributions, our approach shows appealing and better stylization performance than the concurrent stylization approaches. In our future work, we plan to extend it for video input.

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
