# OpenReview forum: "STAM: Zero-shot Style Transfer using Diffusion Model via Attention Modulation"
_thecvf.com/CVPR/2025/Workshop/CVEU — CVPR 2025_

### Official Review · Reviewer_5BZG · 2025-03-16
**review of 31**

**Rating:** 2
**Confidence:** 4

**Review:**

This paper proposes a STAM, a zero-shot style transfer with attention modulation. The key idea is an attention modulation that rescale the QKV to emphasize the style impact, and an aggregated attention to regulate the impact of the content and style from the attention-swapping pathways. Experimental results show that the proposed method achieved a better balance between the content and the style. However, the presentation is poor, with many unclear or misleading descriptions. For example, in Line 320, `this naïve method` is not the method using Eq. (6), based on my understanding. How it is implemented should be explicitly given. Fig. 2. and Fig. 4 are difficult to follow since it is different variants of the baselines are not clearly introduced in the Sec. 3.1. BTW, there should be Sec. 3.2 at Line 297, which is missing. The notation is messy. For example, in Line 491 CQs, CKs, and CVs are introduced but are not used in the method. Overall, the paper is hard to follow, which makes it not ready for publication.

Other issues:
1. The related work section is messy. For example, StyleDiffusion is categized into both training-based (Line 166) and fine-tuning-based (Line 174). Line 191, `however, these methods struggle with contextual fidelity—e.g., StylAligned distorts content—prompting InstantStyle to integrate pre-trained networks for consistency` is very hard to understand. `these methods` are MasaCtrl and PnP, why suddenly referring to StylAligned and InstantStyle? In addition, the related work part is suggested to analyze the difference between the proposed method and the introduced related work.
2. All equations: Q,K should be QK. There is no comma in-between
3. U-Net and UNet should be consistent
4. It is unclear why 0.5 is used in Eq. (10)
5. Did the running time of DDIM inversion is included in Table 1?

---

### Official Review · Reviewer_cLtu · 2025-03-20
**A new attention modulation method to achieve zero-shot style transfer**

**Rating:** 4
**Confidence:** 4

**Review:**

**Strength**

++ The paper proposes a novel attention modulation and attention aggregation method for zero-shot style transfer.

++ The method is free of additional adapters and just utilizes a pre-trained diffusion model.

++ Paper is well written and the comparisons with previous methods are extensive.

**Weakness**

-- In some results, the identity and details of the content image seems to be changed, such as the house case in Figure 7. The authors also discussed the tradeoff of style transfer v.s. content preservation in Figure 10. Therefore, choosing the hyperparameter for attention modulation is important and may be influenced by different style images.

---

### Official Review · Reviewer_XTjC · 2025-03-23
**STAM: Zero-shot Style Transfer using Diffusion Model via Attention Modulation**

**Rating:** 4
**Confidence:** 4

**Review:**

This paper proposes a zero-shot image stylization approach that achieves fine control between style regulation and content preservation through a novel attention modulation and aggregation scheme, enabling style regulation without latent tuning, preserving content information, and addressing prompt-image misalignment without additional adapters. However, several concerns and questions remain regarding this paper.

1.	This paper utilizes attention control to perform the style transfer task. I would like to know how the method in this paper differs from the two papers presented at CVPR 2024. One is Zero-shot Style Transfer via Attention Rearrangement, and the other is Style Injection in Diffusion: A Training-free Approach for Adapting Large-scale Diffusion Models for Style Transfer. Could you compare these methods and highlight the key differences?
2.	I suggest that the authors include some quantitative experiments in the ablation study, rather than relying solely on generated images for comparison. This would provide a more comprehensive and thorough evaluation.

---

### Official Review · Reviewer_CJ9b · 2025-03-25
**Review for STAM, a zero shot style transfer method**

**Rating:** 4
**Confidence:** 4

**Review:**

Summary:
In this paper, the authors propose a method called STAM that attempts the task of style transfer in a zero-shot manner. The  method first does DDIM inversion, and then uses aggregated attention across three branches: content image reconstruction, style image generation, and stylized output image.


Pros:
- the method is zero shot and does not require any additional training.
- the use of attention maps for controlling the style and content look correct and follows the current consensus in the literature.
- the results shown in the paper look compelling.

Cons:
- The writing of the paper is a little difficult to follow, especially the method section.
- Figure 3 is not explained sufficiently. I would recommend the authors explain the three blocks shown in yellow on the right in more details.

---

### Decision · Program_Chairs · 2025-03-25

**Decision:**

Accept

**Comment:**

The paper proposes STAM, a zero-shot style transfer method using novel attention modulation and aggregation mechanisms without additional model training. Reviewers praised the approach for its compelling qualitative results, clear novelty, and extensive comparisons. However, notable concerns were raised regarding clarity and presentation, specifically unclear writing, insufficient explanations of key figures, and messy notation.

Given the clear strengths and valuable contributions identified by most reviewers, the paper is accepted. Authors are strongly advised to thoroughly revise the manuscript's clarity, notation, figure explanations, and address reviewer comments regarding related work comparisons and quantitative ablations in the camera-ready version.